# Antioxidant Capacity of Herzegovinian Wildflowers Evaluated by UV–VIS and Cyclic Voltammetry Analysis

**DOI:** 10.3390/molecules27175466

**Published:** 2022-08-25

**Authors:** Gloria Zlatić, Anamarija Arapović, Ivana Martinović, Anita Martinović Bevanda, Perica Bošković, Ante Prkić, Andrea Paut, Tina Vukušić

**Affiliations:** 1Department of Chemistry, Faculty of Science and Education, University of Mostar, 88000 Mostar, Bosnia and Herzegovina; 2Department of Chemistry, Faculty of Science, 21000 Split, Croatia; 3Department of Chemistry, Faculty of Chemistry and Technology, 21000 Split, Croatia

**Keywords:** cyclic voltammetry, spectrophotometry, plant extracts, antioxidant composite index, phenols, FRAP, ABTS

## Abstract

Considering the vast cultural and traditional heritage of the use of aromatic herbs and wildflowers for the treatment of light medical conditions in the Balkans, a comparison of the antioxidant capacity of wildflowers extracts from Herzegovina was studied using both cyclic voltammetry and spectrophotometry. The cyclic voltammograms taken in the potential range between 0 V and 800 mV and scan rate of 100 mV s^−1^ were used for the quantification of the electrochemical properties of polyphenols present in four aqueous plant extracts. Antioxidant capacity expressed as mmoL of gallic acid equivalents per gram of dried weight of the sample (mmoL GAE g^−1^ dw) was deduced from the area below the major anodic peaks (Q400 pH 6.0, Q500 pH 4.7, Q600 pH 3.6). The results of electrochemical measurements suggest that the major contributors of antioxidant properties of examined plants are polyphenolic compounds that contain ortho-dihydroxy-phenol or gallate groups. Using Ferric reducing-antioxidant power (FRAP) and 2,2′-azino-bis spectrophotometric methods (3-ethylbenzthiazoline-6-sulphonic acid) radical cation-scavenging activity (ABTS) additionally determined antioxidant capacity. The FRAP results ranged from 2.9702–9.9418 mmoL Fe/g dw, while the results for ABTS assays expressed as Trolox equivalents (TE) ranged from 14.1842–42.6217 mmoL TE/g dw. The Folin–Ciocalteu procedure was applied to determine the total phenolics content (TP). The TP content expressed as Gallic acid equivalents (GAE) ranged from 6.0343–9.472 mmoL GAE/g dw. The measurements of total flavonoid (TF) and total condensed tannin (TT) contents were also performed to obtain a broader polyphenolic profile of tested plant materials. *Origanum vulgare* L. scored the highest on each test, with the exception of TT content, followed by the *Mentha × piperita* L., *Artemisia annua* L., and *Artemisia absinthium* L., respectively. The highest TT content, expressed as mg of (−)catechin equivalents per gram of dried weight of sample (mg CE/g dw), was achieved with *A. absinthium* extract (119.230 mg CE/g dw) followed by *O. vulgare* (90.384 mg CE/g dw), *A. annua* (86.538 mg CE/g dw) and *M. piperita* (69.231 mg CE/g dw), respectively. In addition, a very good correlation between electrochemical and spectroscopic methods was achieved.

## 1. Introduction

Oxidative stress is the result of an imbalance of oxidative and reducing species in organisms and represents a disruption of the redox system within signal transudation pathways [1,2]. Although environmental stressors (i.e., UV, ionizing radiations, pollutants, and heavy metals) can influence the cell physiology, oxygen reactive species (ROS) accumulated through oxygen metabolism coupled with the inability of organisms to detoxify can seriously damage the biological systems [3,4,5,6,7,8,9,10,11]. ROS are normally present in the physiological solutions of functional cells. If certain levels are exceeded, these reactive oxygen species can attack vital biomolecules such as nucleic acids, lipids, proteins, polyunsaturated fatty acids, or carbohydrates. DNA damage is linked with the pathophysiology of diseases, such as atherosclerosis, type 2 diabetes, cardiac impairment, Alzheimer’s, Parkinson’s, and cancer [12].

Plants are a precious source of polyphenolic compounds that exhibit excellent antioxidant activity and can inhibit the oxidation of polyunsaturated fatty acids by free radicals in membranes of human body cells [13,14,15,16,17,18,19,20,21]. The antioxidative capacity of human cells is dependent on the enzymatic mechanisms, but the external supply of natural antioxidants contributes to the antioxidant capacity of blood plasma and is important for human health. These compounds convert rapidly and should be included in the daily diet [22,23]. Stress is known to alter the human approach to food and medication sources [24,25,26,27]. A recent unofficial study suggests that adolescents in Bosnia and Herzegovina have changed their eating habits during the COVID-19 pandemic and it is reported that there has been an increase in the use of dietary supplements in the tested population. [28]

Tea infusions of dried fruits and herbs are often consumed as beverages in many parts of the world. The sub-Mediterranean climate of the Southern region of Bosnia and Herzegovina conditioned the development of vast biodiversity in aromatic and medicinal plants which was recognized by locals and led to a long tradition—first in a collection of medicinal and aromatic wildflowers and then its cultivation for use in treating light health conditions [29]. The cultural and traditional heritage of the use of aromatic herbs and wildflowers for the treatment of light medical conditions remains for most of the Balkans, including Kosovo [30], Bulgaria, [31] North Macedonia [32], Serbia [33,34], Croatia [35], Albania [36] and Montenegro [37]. The mentioned area has a population of approximately 50 million people. Approximately 160 autochthonous aromatic and medicinal plants are collected or produced only in Croatia. [38] The aerial parts of plants are collected in season, dried and served topped with hot water as an alternative remedy or used in the local production of alcoholic drinks, such as “travarica” or “rakija” [39,40].

Despite the cultural and biological diversity of this region, folk medicine and their associated traditions do not have a role in the rural economic development of Bosnia and Herzegovina, yet the extensive exploitation and illegal export of medicinal plants are present [41]. Medicinal plants like *Mentha piperita* are often used in the treatment of light health conditions such as gastrointestinal ailments, respiratory and dermatologic problems; however, ethnobotanical studies also reported that medicinal plant species previously had a much broader spectrum of application, and that the tradition of collecting and consuming dried wildflowers is prevalent in the rural areas mostly within the elderly population and during a food shortage [24,42,43,44,45]. Common wormwood, wild marjoram, and peppermint help the treatment of digestive problems, while sweet wormwood can boost the immune system [46], which can be a good alternative to the use of expensive medicine [47,48].

These plants can be found in herbaria of Balkans countries (*A. annua*; BEOU, SARA: 42554−42562, *A. absinthium*; BEOU, SARA: 42539−42552, 51123, 51124, 51702, NMNHS-BOT: 000000000567, 000000000569, FCD: 201912707, 18791256, 19944, *O. vulgare*; BEOU, SARA: 34280−34349, 49817, 51597, 52302, NMNHS-BOT: 000000000349, 000000000350, *M. piperita*; FCD [49], ZAGR: 57947 [50]), where all taxa are native to the Balkans, except for *M. piperita*, which was cultivated in these areas [51]. Although herbal teas are popular beverages, there is a growing interest in creating safe and non-toxic corrosion inhibitors of metals and alloys from the herbal extract [52,53,54,55,56]. The antioxidant properties are often examined by UV–Visible and ESR spectrometry, chromatography, fluorescence, chemiluminescence, and electrochemistry. Electrochemical methods are increasingly used as a rapid, precise, accurate, and economic test for polyphenolic content [57,58,59,60,61,62,63,64,65,66].

The aim of the study was to assess the polyphenolic content and antioxidant properties of wildflowers commonly used in households across Herzegovina with a simple method without expensive equipment, since the interest in monitoring or advertising the quality of native aromatic and medicinal wildflowers in Bosnia and Herzegovina is fading. The electrochemical behaviors of the antioxidants present in sweet and common wormwood (*Artemisia annua* L., *Artemisia absinthium* L.), wild marjoram (*Origanum vulgare* L.), and peppermint (*Mentha × piperita* L.) aqueous extracts were assessed at different pH to quickly and efficiently evaluate the antioxidant capacity of wildflowers. The obtained cyclic voltammograms were compared with the voltammograms of standard polyphenolic compounds and the results of spectrophotometric methods. The total antioxidant capacity of four different plant extracts was measured using three in vitro methods: cyclic voltammetry (CV), ABTS radical cation scavenging activity, ferric reducing-antioxidant power (FRAP). The content of total phenolics (TP) in examined aqueous plant extracts was assessed by the Folin–Ciocalteu (FC) reaction, which is based on the measurement of the reductive capacity of antioxidants present in the examined samples. To acquire a broader polyphenolic profile of tested wildflowers, total flavonoid (TF), and the total condensed tannins (TT) content was also determined.

## 2. Results and Discussion

Electrochemical experiments were carried out in three different buffer solutions: potassium phosphate buffer, pH 6.0; sodium citrate, pH 4.7; and sodium acetate, pH 3.6, to examine the effect of pH change on ACI. To cover all groups of antioxidant compounds, cyclic voltammograms were recorded in the potential range of 0–800 mV, at a scan rate of 100 mV s^−1^ (Figure 1). Nikolić et al., provided the oxidation potentials of some phenolics at pH 4.5. Quercetin has two oxidation peaks that appear at 0.320 V and 0.537 V, respectively. Caffeic acid shows an oxidation peak at 0.344 V, while the kaempferol, ferulic acid, and coumaric acid have oxidation peaks in the sodium acetate buffer of pH 4.5 at 0.548, 0.604, and 0.759 V, respectively [61].

Figure 1 shows cyclic voltammograms of 40 mg L^−1^ of gallic acid, ascorbic acid, (−)catechin, and caffeic acid in a sodium acetate buffer solution at pH 3.6 recorded at the scan rate of 100 mV s^−1^. Ascorbic acid showed one irreversible oxidation peak at 0.200 V, while the gallic acid showed an irreversible oxidation peak at 0.407 V. Reversible oxidation peaks were observed for (−)catechin and caffeic acid solution. The first oxidation peaks for (−)catechin and caffeic acid were observed at 0.400 V and 0.441 V, respectively. The second oxidation peak was observed for (−)catechin at 0.722 V.

As shown in Figure 2b, samples S1 and S2 show an oxidation peak of approximately 0.340 and 0.360 V at pH 4.7, respectively. This oxidation peak could correspond to the oxidation of caffeic acid that has been earlier identified in tea extract of *A. annua* [67]. Additionally, cathodic peaks were observed at 0.305 V (S1) and 0.295 V (S2), which is in agreement with previous studies for caffeic acid that reported a reduction peak potential at 0.273 V [61]. Reversible oxidation peaks were also observed in the cyclic voltammograms of S1 (0.447 V) and S2 (0.429) recorded in sodium acetate buffer solution, pH 3.6 (Figure 2c). 

The mechanism of oxidation of caffeic acid has been studied and it is accepted that oxidation of caffeic acid leads to o-quinones via semiquinone forms [68,69]. The potential at which the second oxidation peak appears for sample samples S1, S2, and S4 at pH 4.7 is approximately 0.590 V, which may be attributed to the oxidation of kaempferol, which was identified earlier in samples S2 and S4 [70,71]. Reversible cathodic peaks also appeared at approximately 0.247 V in cyclic voltammograms of samples S3 and S4. Voltammograms of samples S3 and S4 exhibited high current oxidation and reduction peaks at all three buffer solutions and suggest that the product of electrochemical oxidation is reversible (Figure 2). 

Table 1 shows the potentials at which oxidation processes are occurring along with the corresponding currents, charges and calculated GAE values for cyclic voltammograms of examined aqueous plant extracts in three buffer solutions (pH 6.0, 4.7, 3.6). Cyclic voltammograms of S3 show two major oxidation peaks at the potential of 0.266 V and 0.342 V (pH 6.0), 0.341 V and 0.417 V (pH 4.7), 0.437 V and 0.517 V (pH 3.6), while the oxidation peaks for S4 appears at 0.239 V (pH 6.0), 0.314 V (pH 4.7), 0.430 V (pH 3.6). The corresponding oxidation peak of reversible redox reaction at pH 3.6 appears at the potential of 0.341 and 0.314 V for samples S3 and S4, respectively. The cathodic peaks of samples S3 and S4 could correspond to a reversible quinone reduction. Quinone is a product of oxidation of the ortho-dihydroxy-phenol and gallate group of phenolics and has a cathodic peak of approximately 0.350 V at pH 3.0. Polyphenolic compounds that contain ortho-dihydroxy-phenol or gallate group have a low formal potential and are readily oxidized at the potential of 0.440 V at pH 3.0 [72]. The second oxidation peak in sample S3 is probably due to oxidation of intermediate species formed after the first electron was donated. HPLC analysis of *M. piperita* (S3) identified catechin among many other polyphenols [73]. Catechin can undergo a quasi-reversible redox reaction due to the meta-diphenol groups on the A-ring and has an oxidation peak of approximately 0.391 V at pH 3.6 [74]. As shown in Figure 2c, the potential of the first oxidation peak for all samples appears between 0.430 and 0.450 V at pH 3.6; therefore, catechin may contribute to the antioxidative capacities of all examined plant samples. Additionally, cyclic voltammograms of 40 mg L^−1^ (−)catechin and caffeic acid in sodium acetate buffer solution recorded at pH 3.6 showed a reversible oxidation peak at 0.400 V and 0.441 V, respectively. But it should also be considered that oxidation of hydroxyl on the C-ring on the gallate group of phenolics is energetically more favorable than oxidation of meta-diphenol groups on the A-ring of catechin [75,76,77].

The potential at which the anodic oxidation peak of sample S4 appears in a more acidic buffer solution (pH 3.6), moves to more positive values (Figure 2d). It is also visible that with the increase in pH value, the potential at which the oxidation peak is observed moves to more negative values. As the pH decreases, the potential of the first oxidation peak for all examined plant extracts increases. Calculated GAE values of samples at pH 3.6 range from 1.22 to 7.94 mg GA g^−1^ dw, while for the pH 4.7 and 6.0 range from 0.49 to 6.57 mg GA g^−1^ dw and 0.79 to 6.33 mg GA g^−1^ dw, respectively. The highest values of GAE are reached at the lowest pH for all examined samples. Phenolic acids and flavonoids increase radical scavenging ability when deprotonated because electron donation becomes easier. The ionization potential of the hydroxyflavones (HF) becomes significantly lower with a lower pH value because, with the deprotonation of HF, electron donation becomes a dominant mechanism of antioxidant action [70]. In addition, it can be seen from Table 1 that antioxidant composite index (ACI) varies in different buffer solutions. Nevertheless, sample S3, *O. vulgare* scored highest on ACI index in all three buffer solutions, followed by sample S4, *M. piperita*, then sample S1, *A. annua* and the lowest ACI index belongs to sample S2, *A. absinthium* which is in accordance with the spectrophotometric analysis.

Antioxidant capacities cannot be completely described by one method. In our study, spectrophotometric techniques such as FRAP and ABTS were used to determine the antioxidant capacity of aqueous plant extracts. The final results of antioxidant capacities of examined herbs are summarized in Table 2. FRAP activity ranged from 2.9702 mmoL Fe/g dw to 9.9418 mmoL Fe/g dw. The obtained results for ABTS ranged from 14.1842 mmoL TE/g dw to 42.6217 mmoL TE/g dw. In both assays, the highest antioxidative activity was found in *Origanum vulgare* (9.95 mmoL Fe/g dw for the FRAP assay and 42.6217 mmoL TE/g dw for ABTS assay). Teas are known to have a high content of polyphenolics and it is known that total polyphenols have a strong correlation with antioxidant capacity. Previous studies of the antioxidant properties of aqueous extracts of the selected plants (S1–S4) reported TP content of 39.58  ±  6.01 mg GAE/g for *A. annua* [78], 58.66 ± 2.16 μg GAE/mg dw [79] and 134.47 mg GAE/100 g dw [80] for *A. abisinhtium*, 10.29 ± 0.04 mg GAE/g dw for *O. vulgare* [81] and 230.8 mg GAE/g dw for *M. pipperita* [82]. As it can be seen from Table 2, higher TP content was recorded for plants harvested and processed in Herzegovina, which varied from 6.0343 mmoL GAE/g dw to 9.472 mmoL GAE/g dw. This could be due to the different physical conditions during growth stages, as well as during the collecting, processing, and storing the plant materials. 

The highest total polyphenolic content among the tested plant extracts was found in the *O. vulgare* sample (9.472, mmoL GAE/g dw). The same trend was observed for the TF content; the highest value was observed for S3, followed by S4, S1 and S2, respectively. As shown in Table 2, the TF varied from 13.923 to 253.191 mg CE/g dw, while the TT content varied from 69.231 to 119.230 mg CE/g dw. The highest TT content was recorded for S2, *A. absinthium.*

All examined samples exhibited antioxidant activity, so in order to compare the measured antioxidant capacities, the ACI indices of the antioxidant capacity scaled to the relative percentages were obtained from the FRAP, ABTS, and CV results. An index value of 100 was assigned to the sample that scored highest on FRAP, ABTS and CV measurements (Table 3). The ACI was then calculated as mean value of individual scores of ABTS, FRAP and CV indices scaled to relative percentages. As shown in Table 3, S3 samples (*O. vulgare* samples) scored the highest ACI, followed by S4 (*M. piperita*), S1 (*A. annua*) and S2 and (*A. absinthium*), which is in accordance with their antioxidant capacity and total phenols.

In electrochemical measurements, the calculated GAE values were less than values obtained with FC assays. In the spectrophotometric assays performed in this study, the sample and reagent are mixed and set aside for the redox reaction (electron transfer) to occur. During the cyclic voltammetry, the polyphenolic compounds are rapidly deprotonated, which means only the polyphenols that are readily deprotonated at 0–800 mV potential range are oxidized. Nevertheless, we have obtained a very good correlation between electrochemical and spectroscopic methods. In all methods, with the exception of TT content, the antioxidant properties of *Origanum vulgare* samples were the highest, followed by the *M. piperita*, *A. annua*, *A. absinthium* samples. The correlation between the FRAP and CV results (Figure 3a) could not be considered statistically significant because of the small *R*^2^ values and *p* > 0.05 (*R*^2^ = 0.8932, *p* = 0.0549; *R*^2^ = 0.8734, *p* = 0.0654; *R*^2^ = 0.8227, *p* = 0.0930). On the contrary, positive correlations were confirmed between ABTS and CV measurements (*R*^2^ = 0.9770, *p* = 0.0115; *R*^2^ = 0.9895, *p* = 0.0053; *R*^2^ = 0.9752, *p* = 0.0125) (Figure 3b). Furthermore, correlations were also found between total phenols and CV assays (*R*^2^ = 0.9183, *p* = 0.0417; *R*^2^ = 0.9260, *p* = 0.0377; *R*^2^ = 0.9551, *p* = 0.0227) (Figure 3c).

## 3. Materials and Methods

### 3.1. Reagents

Gallic acid, 2,4,6-tris(2-pyridyl)-s-triazine (TPTZ), sodium acetate, iron (II) sulphate heptahydrate, 2,2′-azinobis-3-ethylbenzothiazoline-6-sulfonate (ABTS), potassium peroxodisulfate, 6-hydroxy-2,5,7,8-tetramethylchroman-2-carboxylic acid (Trolox), ascorbic acid, (−)catechin and caffeic acid were purchased from Sigma-Aldrich (Steinheim, Germany). Sulphuric acid, ethanol, hydrochloric acid, iron(III) chloride hexahydrate, acetic acid, sodium hydroxide and methanol were purchased from (Honeywell, Charlotte, NC, USA). Folin–Ciocalteu’s reagent was purchased from Carlo Erba (Chau. du Vexin, France). Citric acid and aluminum chloride were purchased from Merck Group (Darmstadt, Germany). Sodium citrate, sodium nitrite and vanillin were purchased from Semikem (Sarajevo, BiH). Potassium dihydrogen phosphate and di-potassium hydrogen phosphate were purchased from Kemika (Zagreb, Croatia). All standard solutions, electrolytes, and sample solutions were prepared with ultrapure water (0.05 μS cm^−1^, Millipore Simplicity UV Water Purification System, Millipore, Burlington, MA, USA) using analytical-reagent grade chemicals.

### 3.2. Sample Preparation

Four different herbs were used as sample tea infusions for beverage consumption: sweet and common wormwood (*Artemisia annua* L., S1, *Artemisia absinthium* L., S2), wild marjoram (*Origanum vulgare* L., S3) and peppermint (*Mentha × piperita* L., S4). Plant materials (series 25/21) were donated by Vextra d.o.o. (Dr. Ante Starčevića 38, Mostar, Bosnia & Herzegovina). Aerial parts of the plants were collected in Herzegovina in their natural habitat in 2021. The examined material was collected, prepared, and stored according to Ph. Eur. 5 [83]. Plants S1, S3 and S4 were collected in the flowering stage, while the S2 was collected right before its flowering. Botanical determination was performed by Mr. Sc. Zdravko Rajić, pharmacist and founder of Vextra doo, after which the authentication was verified using The Plant List database (source TICA for S1 and S2, source WCSP for S3 and S4) [84]. All plant material was air-dried at temperatures below 40 °C and kept in the dark. The samples of aqueous herbal extracts were prepared by immersing 2.0000 ± 0.0001 g dried plants in 200 mL of 95 °C ultra pure water for 10 min. Before analysis, extracts were filtered through a 0.45 μm filter paper and diluted up to 250 mL. For spectrophotometric measurements, samples were diluted to 0.2 g L^−1^.

### 3.3. Instruments

All electrochemical measurements were performed with an Autolab PGSTAT320N (Metrohm Autolab B.V., Utrecht, The Netherlands) controlled by Nova software, Version 1.5.018. The pH of buffer solutions used in electrochemical measurements was measured with WTW inoLab pH meter (pH 7110, Xylem Analytics Germany GmbH, Weilheim, Germany) equipped with a glass electrode. For the absorbance measurements, an UV-1800, UV–Visible Spectrophotometer (Shimadzu Corporation, Kyoto, Japan) with paired quartz cuvettes (optical path 1 cm) was used.

### 3.4. Electrochemical Determination of Antioxidant Capacity

The electrochemical measurements were performed in a standard three-electrode cell. The counter electrode was a platinum electrode and the reference electrode, to which all measured potentials are referred, was an Ag |AgCl| 3 M KCl. A graphite electrode, 6 mm in diameter, was used as a working electrode. Prior to each measurement, the electrode was abraded by using silicon carbide papers from 150 to 1000 grit, then washed with distilled water, ultrasonically degreased in ethanol, and finally dried. Experiments were performed in the three buffer solutions: 0.3 mol L^−1^ sodium acetate buffer pH 3.6, 0.1 mol L^−1^ sodium citrate buffer pH 4.7, 0.2 mol L^−1^ potassium phosphate buffer pH 6.0. The supporting electrolyte in phosphate buffer was 0.1 mol L^−1^ potassium chloride. The electrochemical behavior of antioxidants in 50 mL buffer solutions (pH 3.6, 4.7, 6.0), with the addition of 10 mL of samples (S1, S2, S3, S4) was studied by cyclic voltammetry (CV) in the potential range between 0 V and 800 mV and scan rate of 100 mV s^−1^. Cyclic voltammograms were also recorded for Gallic acid in the concentration range (2.4–82.0 mg L^−1^). A calibration curves (y = 0.5525x − 0.7323, *R*^2^ = 0.9985 for pH 6.0, y = 0.5283x + 0.4802, *R*^2^ = 0.9929 for pH 4.7, and y = 0.5134x + 2.2963, *R*^2^ = 0.9924 for pH 3.6) were obtained from the area below the major anodic peaks (Q400 pH 6.0, Q500 pH 4.7, Q600 pH 3.6) vs. concentration, c, to calculate mg of gallic acid equivalents (GAE) per g of dried weight (dw) of studied plants. Cyclic voltammograms were also recorded severally for 40 mg L^−1^ solutions of gallic acid, ascorbic acid, (−)catechin, and caffeic acid in sodium acetate buffer solution pH 3.6 to compare the results with the signals obtained with the samples. All measurements were performed in triplicate.

### 3.5. Antioxidative Assays

Ferric reducing-antioxidant power (FRAP) was measured using the method introduced by Benzie and Strain (1996) [85]. The working FRAP reagent was freshly prepared prior to each measurement by mixing 2.5 mL of TPTZ solution with 2.5 mL FeCl_3_·6H_2_O and 25 mL sodium acetate buffer solutions (pH 3.6).

To obtain the standard calibration curve, the absorbance of FeSO_4_·7H_2_O (0.05–0.8 mmoL L^−1^) solutions was measured at 593 nm. 2.0 mL of diluted samples were mixed with 1.8 mL of FRAP working reagent. After 10 min of incubation at 37 °C, the absorbance was measured at 593 nm in triplicates. The final FRAP results were calculated from the calibration curve y = 2.1755x − 0.0439, *R*^2^ = 0.9998 and were expressed as mmoL Fe per gram of dry weight of plant sample (mmoL Fe/g dw).

ABTS assay was executed as described by Re et al., with slight modifications [86]. Stock solution 1 (ABTS, 7 mmoL L^−1^) and stock solution 2 (potassium persulfate, 2.45 mmoL L^−1^) were prepared in sodium acetate buffer, pH 4.5 and kept in dark (0–4 °C). Working ABTS solution was prepared by mixing 5 mL of stock solution 1 and 2, after which the working solution was kept in the dark for 12–16 h. Finally, the ABTS working solution was diluted with sodium acetate buffer (pH 4.5) to obtain the absorbance of 0.74 ± 0.02 at 734 nm. In four milliliters of the ABTS working solution, 2.0 mL of the sample was added, after which the mixture was kept in the dark for 7 min. Finally, the decrease in absorbance was measured at 734 nm. Redistilled water served as the blank control while the Trolox was used as the standard (0.1–1 mmoL L^−1^). The equation of calibration curve was y = −0.608x + 0.7548, *R*^2^ = 0.9645. The results were expressed as millimoles of Trolox equivalents (TE) per gram of dry weight (mmoL TE/g dw).

### 3.6. Total Polyphenolic Content (TP), Total Flavonoid Content (TF), and Total Condensed Tannins (TT)

Total polyphenols in the plant extracts were determined using the Folin–Ciocalteu procedure described by Singleton et al., and Li et al. [87,88]. Gallic acid (0.05–0.7 mmoL L^−1^) was used as the reference standard. Samples (S1, S2, S3 and S4) were mixed with 2 mL of diluted Folin–Ciocalteu reagent in the ratio of 1:10. The absorbance of samples was measured at 750 nm. The calculated TP values were expressed as gallic acid equivalent (GAE)/g dw of samples [88].

TF was determined as described by Zhishen et al. [89]. A total of 8 mL of water and 0.6 mL of 5% sodium nitrite were added to 0.2 mL of sample, followed by the addition of 0.6 mL of 10% aluminum chloride solution after 6 min. At 5 min, 4 mL of 1 mol L^−1^ sodium hydroxide solution was added, and the samples were diluted up to 20 mL. The absorbance was recorded at 510 nm.

The TT measurements method described by Broadhurst et al. [90] was used. Briefly, 6 mL of 4% vanillin solution prepared in methanol was added to 1 mL of sample, followed by the addition of 3 mL concentrated hydrochloric acid solution. The absorbance was recorded after 15 min of incubation at 510 nm. The calibration curve for TF (y = 0.0282x + 0.0222, *R*^2^ = 0.9973) and TT (y = 0.0026x + 0.002, *R*^2^ = 0.9958) determination was obtained with (–)catechin as a standard in the concentration range of 3–30 mg L^−1^. TF and TT are expressed as milligrams of catechin equivalents (CE) per gram of dry weight (mg CE/g dw).

### 3.7. Antioxidant Composite Index (ACI)

The antioxidant index score was calculated as the average of all tests for each sample according to Equation (1) [70]:Antioxidant index score = [(sample score/best score) × 100] (1)

All assays were assigned equal weight, then the best score for each test was assigned an index value of 100, after which the antioxidant potency composite index was calculated for all other samples. An overall antioxidant potency composite index gave an average value of scores of all tests and samples for comparative study [91].

### 3.8. Statistical Analysis

All presented numeric data are means ± standard deviations (SD) for three measurements. Statistical analysis was performed by GraphPad Prism 9, Version 9.0.0 (121) (GraphPad Software, San Diego, CA, USA). Differences at *p* < 0.05 were considered to be statistically significant.

## 4. Conclusions

The antioxidant capacity of four different plant extracts was investigated using electrochemical and spectrophotometric methods: CV, FRAP and ABTS. To obtain a broader profile of the polyphenol composition of tested plant extracts, spectrophotometric analysis of TP, TF and TT content was executed. CV measurements revealed that electron transfer of polyphenols found in studied plant samples is pH-dependent and reversible. As the pH decreases, the potential of the first oxidation peak for all examined plant extracts increases. The highest values of GAE are reached at the lowest pH for all examined samples. The results of electrochemical measurements suggest that the major contributors to the antioxidant capacity of the examined plant samples are polyphenolic compounds that contain ortho-dihydroxy-phenol or gallate groups and are readily oxidized in the potential range used in this study. In spectrophotometric techniques such as FRAP, ABTS, TP and TF, the best values were obtained for *O. vulgare* samples followed by *M. piperita*, *A. annua* and *A. absinthium*. The same trend in the results was observed for the calculated antioxidant potency composite indices. *O. vulgare* samples had the highest ACI. The highest TT content was observed for *A. absinthium*, followed by *O. vulgare*, *A. annua* and *M. piperita*, respectively. A high positive correlation between the results deduced from electrochemical and spectrophotometric analysis indicates that cyclic voltammetry can be a reliable and quick method for determining the antioxidant properties of aqueous plant extracts. Even though the results of this investigation indicate the favorable antioxidant properties of the studied wildflowers, further in vivo research should be considered to fully understand the radical scavenging activity of the polyphenolic compounds found in tea infusions and their effects on human health.

## Figures and Tables

**Figure 1 molecules-27-05466-f001:**
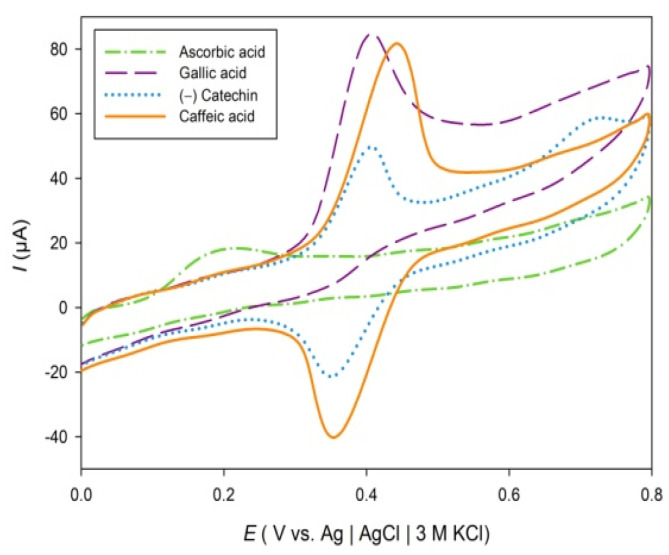
Cyclic voltammograms of 40 mg L^−1^ of gallic acid, ascorbic acid, (−)catechin and caffeic acid in sodium acetate buffer solution pH 3.6 recorded at a scan rate of 100 mV s^−1^.

**Figure 2 molecules-27-05466-f002:**
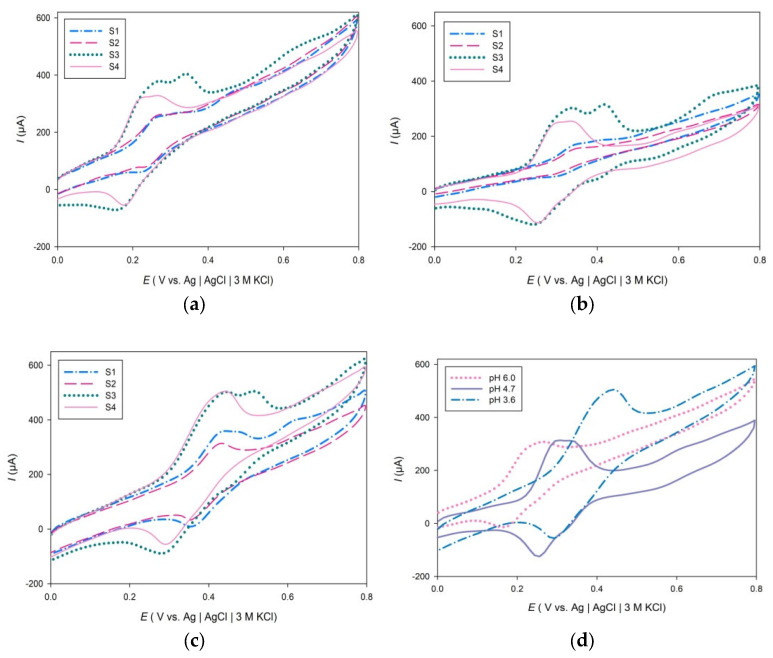
Cyclic voltammograms of aqueous plant extracts (S1−S4) in different buffer solutions pH 6.0 (**a**), pH 4.7 (**b**), pH 3.6 (**c**). Cyclic voltammogram of *M. piperita* (S3) in three buffer solutions (**d**) recorded at a scan rate of 100 mV s^−1^.

**Figure 3 molecules-27-05466-f003:**
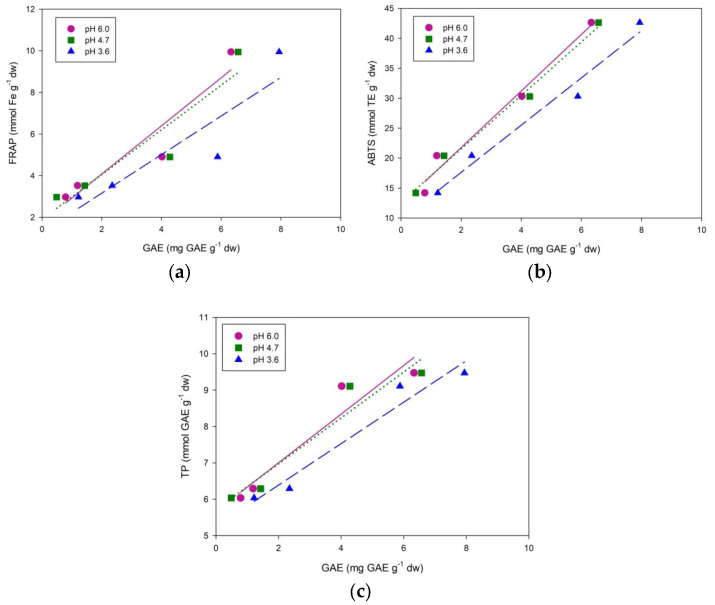
Correlation between cyclic voltammetry (pH 6.0, 4.7 and 3.6) and spectrophotometric results—statistical analysis: (**a**) FRAP (**b**) ABTS (**c**) TP.

**Table 1 molecules-27-05466-t001:** Peak potentials, Ep, and currents, Ip, from cyclic voltammograms of aqueous plant extract of *A. annua* (S1), *A. absinthium* (S2), *O. vulgare* (S3), *M. piperita* (S4) at pH 3.6, 4.7, 6.0.

Herbs	pH	*E*_p_ (V)				*I*_p_, (µA cm^−2^)				*Q* (µ*C*)	GAE (mg GA g^−1^ dw)
		1	2	3	4	1	2	3	4		
S1	6.0	0.288	-	0.471	-	262.52	-	343.51	-	5.18	1.19
S2		0.268	-	-	-	262.63	-	-	-	3.59	0.79
S3		0.266	0.342	-	0.620	384.55	403.12	-	490.27	25.47	6.33
S4		0.239	-	-	-	327.85	-	-	-	12.51	4.02
S1	4.7	0.366	-	0.554	-	343.51	-	233.58	-	6.52	1.43
S2		0.344	-	-	-	154.03	-	-	-	2.55	0.49
S3		0.341	0.417	-	0.696	305.72	315.54	-	356.15	28.26	6.57
S4		0.314	-	-	0.617	256.26	-	-	223.64	18.59	4.28
S1	3.6	0.447	-	0.622	-	358.09	-	400.10	-	11.83	2.35
S2		0.429	-	-	-	313.40	-	-	-	7.56	1.22
S3		0.437	0.517	-	-	498.15	505.39	-	-	32.84	7.94
S4		0.430	-	-	-	500.53	-	-	-	25.11	5.88

**Table 2 molecules-27-05466-t002:** Antioxidant capacity and total phenol, flavonoid and tannin contents in aqueous plant extracts.

Herbs	FRAP mmoL Fe g^−1^ dw	ABTS mmoL TE g^−1^ dw	Total Phenols mmoL GAE g^−1^ dw	Total Flavonoids mg CE g^−1^ dw	Total Tannins mg CE g^−1^ dw
S1	3.5218 ± 0.312	20.4013 ± 0.802	6.2903 ± 0.052	20.922 ± 0.019	86.538 ± 0.046
S2	2.9702 ± 0.168	14.1842 ± 0.294	6.0343 ± 0.089	13.923 ± 0.079	119.230 ± 0.008
S3	9.9418 ± 0.568	42.6217 ± 0.615	9.4720 ± 0.052	253.191 ± 0.015	90.384 ± 0.062
S4	4.9008 ± 0.316	30.3026 ± 0.939	9.1063 ± 0.089	82.978 ± 0.077	69.231 ± 0.013

**Table 3 molecules-27-05466-t003:** Antioxidant potency composite index of aqueous plant extracts from four antioxidant capacity measures scaled to relative percentages.

Herbs	ABTS Index	FRAP Index	Q Index			ACI
			pH 3.6	pH 4.7	pH 6.0	
S1	47.87	35.42	36.04	23.10	20.34	32.5
S2	33.28	29.88	23.04	9.02	14.10	21.9
S3	100	100	100	100	100	100
S4	71.10	49.29	76.46	65.78	49.14	56.3

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
