# Peer review of "Antioxidant Capacity of Herzegovinian Wildflowers Evaluated by UV–VIS and Cyclic Voltammetry Analysis"

_molecules, 2022, doi:10.3390/molecules27175466_

Round 1

Reviewer 1 Report

My main impression is that the paper titled “Antioxidant capacity of Herzegovinian wildflowers evaluated by UV-VIS and cyclic voltammetry analysis”, is overall well written, the style is clear, However a some comments should be arranged as indicated below:

-       In abstract the conclusion is not very well detailed

-       What are the criteria used for the choice of your selected plants

-       How did you identify your selected plant species

-       Is there any specimen deposited in herbarium

-       What is the voucher number of your plant species

-       For the authentification of plants used for your experiments you have to check these plants by using the database “The Plant List” (http://www.theplantlist.org/)

 You have to insert  all these information in methodology section

- The conclusion is too long, you have to be succinct

Author Response

Dear Reviewer,

I am sending corrections according to your suggestions.

Kind Regards,

Perica Bošković

Reviewer 2 Report

Antioxidant Capacity of Herzegovinian Wildflowers Evaluated 2 by UV-VIS and Cyclic Voltammetry Analysis

The authors carried out an interesting study comparing rapid techniques for the determination of antioxidant capacity and some families of compounds. In my opinion the work can be accepted in this journal.

I have a few comments that need to be considered and clarified before being accepted.

The resolution of the figures should be improved.

1.       Line 98. Folin− Ciocalteu is used to determine the content of FT, clarify if Folin− Ciocalteu was used to determine the antioxidant capacity.

2.       The authors should clarify why they omitted to use the DPPH method for antioxidant capacity, instead of ABTS.

3.       Line 172. Check the scientific names, they should be written in italics.

4.       Line 292. It is not clear why the authors only used 40 mg/L of each reagent.

5.       Line 336. Why was catechin used instead of tannic acid for the determination of total tannins? It would be the most appropriate.

6.       In all tables and figures, specify that the determinations are based on dry biomass.

7.       In section 3.2. Specify if the plants were collected with roots, and the whole plant was used to make the infusions, or only the aerial part.

8.       Authors should discuss their results in Table 2 against reports for these same species using the same UV-Vis techniques.

Author Response

(The authors gave the same response as above.)

Reviewer 3 Report

The paper can be interesting but needs some details explanations and corrections.

The electrochemical methods can be one reason to publish the paper, since no HPLC or GC to quantify but needs to be improved.

1. Line 106.......the potential range of 0−l800 mV, at

In the abstract say, 0-800 mV which is more possible, as well as from all paper?

2. Fig. 1 and 2 and description...lines 105-107.

First, appear Fig2 in the text and after Fig1.?? Which one is each: Explain and correct. Begin to be coherent in the text as the figures appear.

Also, the colors aren't easy to understand and better legend must be done with more details in figures 1, 2 and 3.

3. The colors in the graphs aren't completed clear.

4. Line 208. The ACI indices scaled... must be better explained and reasons.

5. Lines 230-236.

The correlations present different p values. Can this explain? Also, the R2 is not so good.

6. Are there calibration curves for electrochemical measurements?

7. The equations of calibration curves to the spectrometric methods must be given.

Author Response

(The authors gave the same response as above.)
